# Diversity of Auger Beetles (Coleoptera: Bostrichidae) in the Mid-Cretaceous Forests with Description of Seven New Species

Andrei A. Legalov [1,2,3,*] and Jiří Háva [4]

1 Institute of Systematics and Ecology of Animals, Siberian Branch, Russian Academy of Sciences, 630091 Novosibirsk, Russia

2 Department of Ecology, Biochemistry and Biotechnology, Altai State University, 656049 Barnaul, Russia

3 Department of Forestry and Landscape Construction, Tomsk State University, 634050 Tomsk, Russia

4 Private Entomological Laboratory & Collection, Rýznerova 37/37, 252 62 Prague, Czech Republic

* Correspondence: fossilweevils@gmail.com

**Abstract:** The diversity and abundance of auger beetles were compared with ecologically similar families of other beetles. It was shown that the ecological niche in dead wood, which in the Paleogene belonged to bark and ambrosia beetles, was occupied by Bostrichidae in the Mesozoic. Seven new species, *Poinarinius aristovi* **sp. nov.**, *P. antonkozlovi* **sp. nov.**, *P. lesnei* **sp. nov.**, *P. perkovskyi* **sp. nov.**, *P. zahradniki* **sp. nov.**, *P. borowskii* **sp. nov.**, and *P. cretaceus* **sp. nov.** from the subfamily Alitrepaninae of the family Bostrichidae are described from mid-Cretaceous Burmese amber. The key to the species of the genus *Poinarinius* Legalov, 2018 is given. The new synonym, *Alitrepanum* Peng, Jiang, Engel & Wang, 2022, **syn. nov.** to *Poinarinius*, was established. A list of the fossil Bostrichidae was compiled.

**Keywords:** Bostrichoidea; Alitrepaninae; new species; Myanmar; Cretaceous





## 1. Introduction

Representatives of the beetles are among the main inhabitants of various forests, where they are associated with both living trees and dead decaying wood [1]. Bostirichidae are a small family of the superfamily Bostrichoidea containing approximately 650 recent species that typically develop in dying and dead wood [2–6]. Therefore, Bostirichidae can be considered as a characteristic forest group and reach the greatest diversity in tropical forests. The earliest find of these beetles was from French amber of the Lower Cretaceous (Albian) [7]. Three species were described from the Cenomanian of Asia (Burmese amber) [8–10]. Auger beetles are also known from late Eocene Baltic amber [11,12], the terminal Eocene of Florissant (USA) [13–15], and early Miocene Dominican and Mexican amber [16,17].

Specimens of Bostrichidae are among the common beetles in Burmese amber. Most specimens belong to the subfamily Alitrepaninae, characterized by a body covered with dense long erect or semierect setae, and long first tarsomere. They have so far only been found in Burmese amber [10]. The purpose of our study was to describe new species and show the role of auger beetles and other beetles in Cretaceous forests.

## 2. Materials and Methods

The amber pieces with the described specimens were obtained from mines [18] in the Hukawng Valley of the state of Kachin (Myanmar). The amber of a probable Cenomanian radiometric age was mined from sedimentary beds, indicating that it had been re-deposited [19]. An araucarian tree, possibly *Agathis*, was the source of the amber [20].

The studied specimens were deposited at the Institute of Systematics and Ecology of Animals, Siberian Branch, Russian Academy of Sciences, Novosibirsk (ISEA) and Jiří Háva Private Entomological Laboratory and Collection, Únětice u Prahy, Prague-West, Czech Republic (JHAC).

The diagrams "Number of auger, bark, and ambrosia beetles in the fossil record" and "Species diversity of the auger, bark, and ambrosia beetles in the fossil record" are based on the data of the presented work (Bostrichidae) as well as the literature data on bark and ambrosia beetles [21–38].

Observations and photographs were made with a Zeiss Stemi-2000 stereoscopic microscope with an AxioCam MRc5 camera.

The morphological terminology used in this paper follows Lawrence et al. [39].

Nomenclatural acts introduced in the present work are registered in ZooBank (www. zoobank.org) under LSID urn:lsid:zoobank.org:pub:41E5F39B-2895-4C2C-86E3-B0B5ACBDEE2F.

### 3. Results

*Systematic Paleontology*

Superfamily **Bostrichoidea** Latreitte, 1802.
Family **Bostrichidae** Latreille, 1802.
Subfamily **Alitrepaninae** Peng, Jiang, Engel & Wang, 2022.
Genus *Poinarinius* Legalov, 2018.
Type species. *Poinarinius burmaensis* Legalov, 2018.
*Alitrepanum* Peng, Jiang, Engel & Wang, 2022, **syn. nov.**
Type species. *Alitrepanum aladelicatum* Peng, Jiang, Engel & Wang, 2022.

**Diagnosis**. Body subcylindrical, usually covered with long erect or semierect setae; length 1.7–4.9 mm. Head hypognathous, visible from above, wider than anterior margin of pronotum. Mandible quite large, curved. Galea with dense, long, erect setae. Labrum almost semicircular, with dense setae. Eyes convex, finely faceted. Frons convex or impressed in middle, densely or finely punctate, sometimes transversally-rugose, usually covered with more or less dense erect setae, sometimes with tubercle in the middle. Vertex weakly convex or weakly impressed in middle, punctate, or longitudinally-rugose, sometimes with two tubercles. Antennae quite short, 8-segmented. First antennomere long-conical or suboval. Second antennomere conical. Third antennomeres conical or suboval. Fourth antennomere conical, wide-conical, or suboval. Fifth antennomere wide-conical. Antennal club formed with three terminal transverse antennomeres. Pronotum concave at anterior margin. Disk usually weakly convex, punctate, sometimes longitudinally-rugose. Scutellum trapezoidal. Elytra subcylindrical or suboval. Elytral intervals from wide to narrow, flat or convex, wider, subequal or narrow than striae. Elytra lacking distinct elytral declivity and lobes or with more or less distinct elytral declivity and ones. Elytral declivity almost vertical or oblique. Elytra sometimes with two transverse or longitudinal lobes, or two spines before elytral declivity; with longitudinal lobes or double spines on sides of elytral declivity; with on sides of elytral declivity or before one. Metanepisterna quite wide. Metaventrite 2.4–3.2 times as long as the metacoxal cavity length. Procoxae elongate and widely separated. Metatrochanter triangular, partially separating femur from coxa. Profemora distinctly thickened. Meso- and metafemora weakly thickened. Protibiae curved, with sixth spines, lacking spurs. Meso- and metatibiae flattened, with two apical spurs. Mesotibial spurs equal in length, quite long. Metabibiae with long spur, almost equal in length to the first tarsomere and quite short one. Tarsi with five narrow tarsomeres. Protarsi subequal or longer than protibiae. Meso- and metatarsi longer than meso- and metatibiae. Claws free, laminate at base.

**Notes**. The main differences between the genus *Poinarinius* and the genus *Alitrepanum* were the 9- and 10-segmented antennae and the structure of the apices of the elytra. The study of the material showed that *Poinarinius burmaensis* and *Alitrepanum aladelicatum* have 8-segmented antennae, and the structure of the apex of the elytra can be different, from elytra without a elytral declivity to elytra with one and with spines or lobes. Therefore, *Alitrepanum* Peng, Jiang, Engel & Wang, 2022, **syn. nov.** is synonymous with *Poinarinius* Legalov, 2018. The body is covered with more or less dense long erect or semierect setae, 8-segmented antennae with transverse antennal club, concave at anterior margin pronotum,

elongate and widely separated procoxae, curved protibiae, with sixth spines, lacking spurs, meso- and metatibiae with two apical spurs, metabibiae with two spurs unequal in length and protarsi subequal or longer than protibiae are important characteristics of the species of the genus.

　　*Poinarinius aristovi* Legalov & Háva, **sp. n.** (Figure 1A–F).

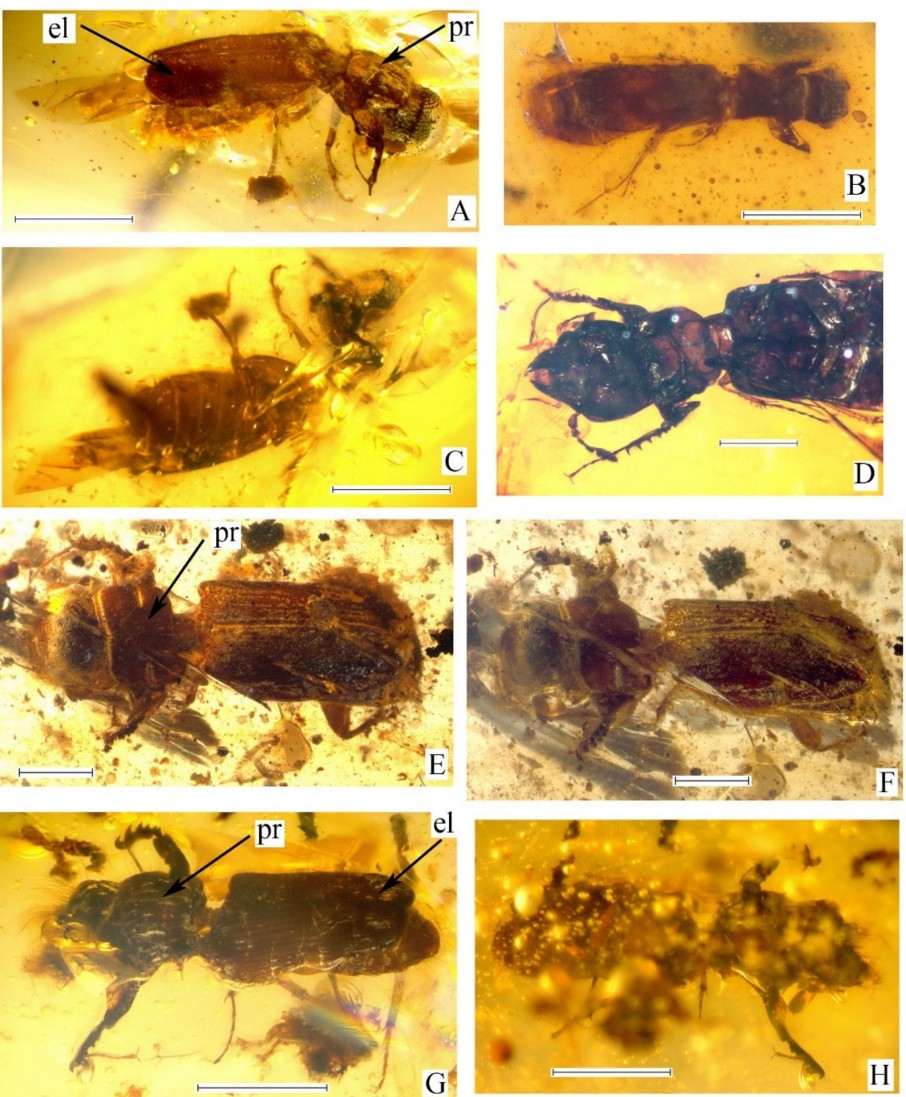

**Figure 1.** *Poinarinius* spp. in Myanmar amber: (**A**) *P. aristovi* sp. nov., holotype, male, ISEA MA2016/2, dorso-lateral view; (**B**) *P. aristovi* sp. nov., paratype, male, ISEA MA 2016/11, dorsal view; (**C**) *P. aristovi* sp. nov., holotype, male, ISEA MA2016/2, ventro-lateral view; (**D**) *P. aristovi* sp. nov., paratype, male, ISEA MA2018/14, ventral view; (**E**) *P. aristovi* sp. nov., paratype, female, ISEA MA2020/27, dorsal view; (**F**) *P. aristovi* sp. nov., paratype, female, ISEA MA2020/27, dorso-lateral view; (**G**) *P. antonkozlovi* sp. nov., holotype, female, ISEA MA2020/24, dorso-lateral view; (**H**) *P. antonkozlovi* sp. nov., holotype, female, ISEA MA2020/24, ventral view. Scale bar 1.0 mm for (**A**–**C**), (**E**–**H**); 0.5 mm for (**D**). Abbreviations: pr—pronotum, el—elytral declivity.

　　LSIDurn:lsid:zoobank.org:act:396688E6-E5AA-48E5-8645-28C614316B24.

　　**Description**. Male. Body brown, length 2.4–3.7 mm. Integument covered with sparse, long, erect setae. Head spherical, densely punctate, 1.1–1.5 times as wide as pronotum apex width. Frons impressed in middle, finely punctate, with setae. Pronotum 0.7–1.6 times as long as wide at apex, 0.6–1.8 times as long as wide in middle, 0.7–1.6 times as long as wide at base. Disk weakly convex, finely and sparsely. Sides weakly rounded, lacking erect setae. Elytra suboval, 2.0–3.4 times as long as the pronotum, 1.9–2.6 times as long as wide

at the base, 1.7–2.3 times as long as wide in the middle, 1.8–2.4 times as long as the apical fourth. Sides of elytra without erect setae. Elytral intervals wide, flat, 3.5–4.0 times wide as striae. Striae not deep, weak. Elytral declivity absent. Metaventrite about 2.4 times as the long as the metacoxal cavity length. First ventrite about 1.4 times as long as the metacoxal cavity length. Second ventrite about 0.9 times as long as the first ventrite. Third ventrite about 1.1 times as long as the second ventrite. Fourth ventrite is 1.0–1.2 times as long as the third ventrite. Fifth ventrite 0.7–1.1 times as long as the fourth ventrite. Metabibial long spur slightly shorter than the fifth tarsomere. Metabibial short, about 0.6 times as long as the fifth tarsomere. Protarsi about 1.5 times as long as the protibiae. Mesotarsi slightly longer than the mesotibiae. Metatarsi about 1.3 times as long as the metatibiae. Protarsi: first tarsomere about 2.7 times as long as the second tarsomere; second tarsomere about 1.2 times as long as the third tarsomere; fourth tarsomere about 0.7 times as long as the third tarsomere; fifth tarsomere about 2.6 times as long as the fourth tarsomere, slightly longer than the third and fourth tarsomeres combined. Mesotarsi: first tarsomere about 1.8 times as long as the second tarsomere; second tarsomere about 1.2 times as long as the third tarsomere; fourth tarsomere subequal to the third tarsomere; fifth tarsomere about 2.4 times as long as the fourth tarsomere, about 1.2 times as long as the third and fourth tarsomeres combined. Metatarsi: first tarsomere about 4.7 times as long as the second tarsomere; second tarsomere subequal to the third tarsomere; fourth tarsomere about 0.9 times as long as the third tarsomere; fifth tarsomere about 2.3 times as long as the fourth tarsomere, slightly longer than the third and fourth tarsomeres combined. Female. Body length 2.6–4.9 mm. Integument covered with long erect setae. Head spherical, densely punctate, about 1.2 times as wide as the pronotum apex width. Frons weakly impressed in middle, with dense erect setae. First antennomere oval, about 1.6 times as long as wide in the middle. Second antennomere conical, about 0.9 times as long as wide at the apex, about 1.3 times as long as and about 0.7 times as narrow as the first antennomere. Third antennomere long-conical, about 1.7 times as long as wide at the apex, 0.5 times as long as and about 0.7 times as narrow as the second antennomere. Fourth antennomere conical, about 0.9 times as long as wide at apex, about 0.6 times as long as and about 1.1 times as wide as third antennomere. Fifth antennomere wide-conical, about 0.7 times as long as wide at apex, about 0.9 times as long as and about 1.1 times as wide as the fourth antennomere. Sixth antennomere wide-conical, about 0.5 times as long as wide at apex, about 0.8 times as long as and about 1.1 times as wide as the fifth antennomere. Pronotum about 0.7 times as long as wide at the apex, about 0.6 times as long as wide in the middle, about 0.7 times as long as wide at the base. Sides with erect setae. Elytra about 3.3 times as long as the pronotum, about 2.1 times as long as wide at the base, about 1.8 times as long as wide in the middle, about 2.2 times as long as the apical fourth. Sides of elytra with erect setae. Metaventrite 2.4–2.6 times as long as metacoxal cavity length. First ventrite about 1.5 times as long as the metacoxal cavity length. Second ventrite subequal to the first ventrite. Third ventrite 1.0–1.3 times as long as the second ventrite. Fourth ventrite 0.8–1.1 times as long as the third ventrite. Fifth ventrite 1.1–1.4 times as long as the fourth ventrite.

**Material examined**. Holotype—male (ISEA), no. MA2016/2. Paratypes: male (ISEA), no. MA2016/11; male (ISEA), no. MA2018/14; male (ISEA), no. MA2020/11; male (ISEA), no. MA2020/13; male (ISEA), no. MA2020/15; male (ISEA), no. MA2020/21; male (ISEA), no. MA2020/22; male (ISEA), no. MA2020/23; male (ISEA), no. MA2020/26; male (ISEA), no. MA2020/28; male (ISEA), no. MA2020/36; male (ISEA), no. MA2020/37; male (ISEA), no. MA2020/38; female (ISEA), no. MA2020/25; female (ISEA), no. MA2020/27; female (ISEA), no. MA2020/39; female (JHAC), no. JH2022/7; female (JHAC), JH2022/10.

**Etymology.** The species is named after the late paleoentomologist Danil S. Aristov (Moscow).

**Comparison.** The new species differs from other species of the genus (excluding *P. antonkozlovi* **sp. nov.**) in the elytra lacking elytral declivity and lobes. It is distinguished from *P. antonkozlovi* **sp. nov.** in the punctate pronotum.

*Poinarinius antonkozlovi* Legalov & Háva, **sp. n.** (Figure 1G,H).

LSIDurn:lsid:zoobank.org:act: 20F616EF-F921-4EB7-9418-41E5B1284936.

**Description**. Female. Body dark brown, length 2.7–2.9 mm. Head slightly wider than the pronotum apical width. Frons impressed, densely punctate, with dense, long, erect setae. Pronotum about 1.1 times as long as wide at the apex, slightly narrower than wide in the middle, slightly longer than wide at the base. Disk weakly convex, coarse, longitudinally-rugose. Sides weakly rounded, with long erect setae. Elytra subcylindrical, about 2.1 times as long as the pronotum, about 1.7 times as long as wide at the base, about 2.0 times as long as wide in the middle, about 2.4 times as long as the apical fourth. Sides of elytra with very sparse, long, erect setae. Elytral intervals narrow, flattened, punctate, 1.5–2.0 times wide as striae. Striae deep. Fifth interval without lobate convexity. Legs with sparse, long, erect setae. Metabibiae with long spur, which is almost equal in length to the first tarsomere and quite short one. Protarsi subequal in length to protibiae. Mesotarsi about 1.3 times as long as the mesotibiae. Metatarsi about 1.2 times as long as the metatibiae. Tarsi longer than tibiae, with five narrow tarsomeres. Protarsi: first tarsomere about 1.4 times as long as the second tarsomere; second tarsomere about 1.3 times as long as the third tarsomere; fourth tarsomere about 0.8 times as long as the third tarsomere; fifth tarsomere about 2.7 times as long as the fourth tarsomere, slightly shorter than the third and fourth tarsomeres combined. Mesotarsi: first tarsomere about 1.3 times as long as the second tarsomere; second tarsomere about 1.4 times as long as the third tarsomere; fourth tarsomere about 0.6 times as long as the third tarsomere; fifth tarsomere about 1.7 times as long as the fourth tarsomere, about 0.6 times as long as the third and fourth tarsomeres combined. Metatarsi: first tarsomere about 2.2 times as long as the second tarsomere; second tarsomere about 1.1 times as long as the third tarsomere; fourth tarsomere about 0.7 times as long as the third tarsomere; fifth tarsomere about 2.8 times as long as the fourth tarsomere, about 1.1 times as long as the third and fourth tarsomeres combined.

**Material examined**. Holotype—female (ISEA), no. MA2020/24. Paratypes: female (ISEA), no. MA2018/15; female (JHAC), no. JH2022/5.

**Etymology.** In honor of Anton Kozlov (Moscow), who helped us in the study.

**Comparison.** The new is very similar to *P. aristovi* **sp. nov.**, but it is easily distinguished by transverse rugose pronotum.

*Poinarinius perkovskyi* Legalov & Háva, **sp. n.** (Figure 2A).

LSIDurn:lsid:zoobank.org:act:10D44B00-6920-4A28-9198-89E319409CFF.

**Description**. Male. Body brown, length 3.0 mm. Integument almost glabrous. Frons weakly impressed in the middle, finely and sparsely punctate, without setae. First antennomere of club about 3.0 times as wide as the length. Second antennomere of club about 4.3 times as wide as the length. Third antennomere of club about 3.6 times as wide as the length. Pronotal disk weakly convex, finely and sparsely punctate. Sides weakly rounded, lacking erect setae. Elytra subcylindrical, about 2.9 times as long as the pronotum. Sides of elytra without erect setae. Elytral intervals wide, flat, about 3.3 times as wide as the striae. Striae not deep. Elytral declivity almost vertical. Elytra with two spines before elytral declivity. Metaventrite 3.0 times as long as the metacoxal cavity length. Metanepisterna 3.1 times as long as wide in the middle. Abdomen convex. First ventrite about 0.8 times as long as the metacoxal cavity length. Second ventrite equal to the first ventrite. Third ventrite slightly longer than the second ventrite. Fourth ventrite slightly longer than the third ventrite. Fifth ventrite about 1.3 times as long as the fourth ventrite. Protibiae curved, with sixth spines, lacking spurs. Meso- and metatibiae flattened, with two apical spurs. Mesotibial spurs equal in length, quite long. Metabibiae with a long spur and quite a short one. Protarsi about 1.4 times as long as protibiae.

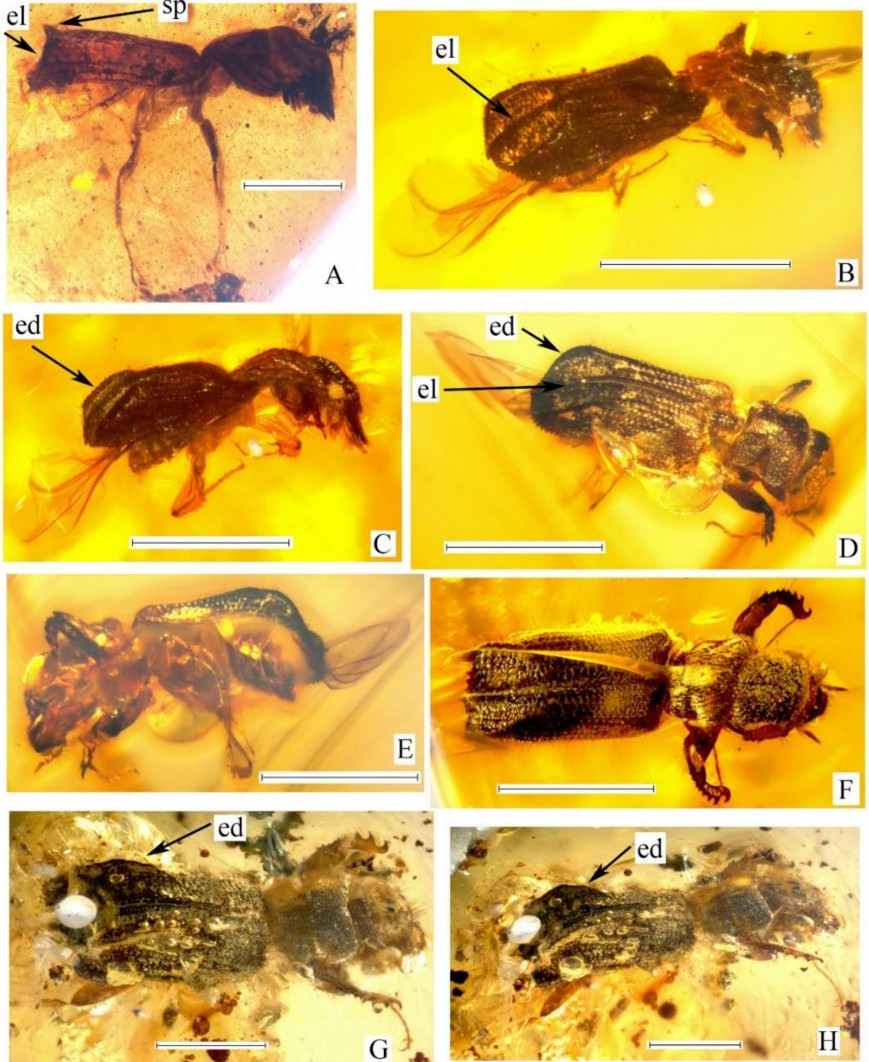

**Figure 2.** *Poinarinius* spp. in Myanmar amber: (**A**) *P. perkovskyi* sp. nov., holotype, male, ISEA MA2020/4, lateral view; (**B**) *P. aladelicatus*, female, ISEA MA 2018/24, dorso-lateral view; (**C**) *P. aladelicatus*, female, ISEA MA2018/24, lateral view; (**D**) *P. aladelicatus*, male, ISEA MA2018/30, dorso-lateral view; (**E**) *P. aladelicatus*, male, ISEA MA2018/30, ventral view; (**F**) *P. aladelicatus*, male, ISEA MA2019/26, dorsal view; (**G**) *P. zahradniki* sp. nov., holotype, female, ISEA MA2018/27, dorsal view; (**H**) *P. zahradniki* sp. nov., holotype, female, ISEA MA2018/27, dorso-lateral view. Scale bar 1.0 mm. Abbreviations: sp—spine, el—elytral declivity, ed—edges of elytral declivity.

Mesotarsi about 1.9 times as long as the mesotibiae. Metatarsi about 1.6 times as long as the metatibiae. Tarsi longer than tibiae, with five narrow tarsomeres. Protarsi: first tarsomere about 2.3 times as long as the second tarsomere; second tarsomere about 0.8 times as long as the third tarsomere; fourth tarsomere about 0.5 times as long as the third tarsomere; fifth tarsomere about 2.2 times as long as the fourth tarsomere, slightly shorter than the third and fourth tarsomeres combined. Mesotarsi: first tarsomere about 2.0 times as long as the second tarsomere; second tarsomere about 0.9 times as long as the third tarsomere; fourth tarsomere about 0.5 times as long as the third tarsomere; fifth tarsomere about 2.9 times as long as the fourth tarsomere, slightly shorter than the third and fourth tarsomeres combined. Metatarsi: first tarsomere about 2.8 times as long as the second tarsomere; second tarsomere about 0.9 times as long as the third tarsomere; fourth tarsomere about 0.5 times as long as the third tarsomere; fifth tarsomere about 3.7 times as long as the fourth tarsomere, about 1.2 times as long as the third and fourth tarsomeres combined.

**Material examined**. Holotype—male (ISEA), no. MA2020/4.

**Etymology.** In honor of Evgeny E. Perkovsky (Kiev) who contributed to the study of amber faunas of the Cretaceous and the Eocene.

**Comparison.** The new differs from *P. aladelicatus* in almost vertical elytral declivity. It is very similar to *P. lesnei* **sp. nov.**, but is distinguished by the elytra with two spines before elytral declivity.

*Poinarinius aladelicatus* (Peng, Jiang, Engel & Wang, 2022), **comb. nov.** (Figure 2B–F).

*Alitrepanum aladelicatum* Peng, Jiang, Engel & Wang, 2022: 3–4, Figures 1–3.

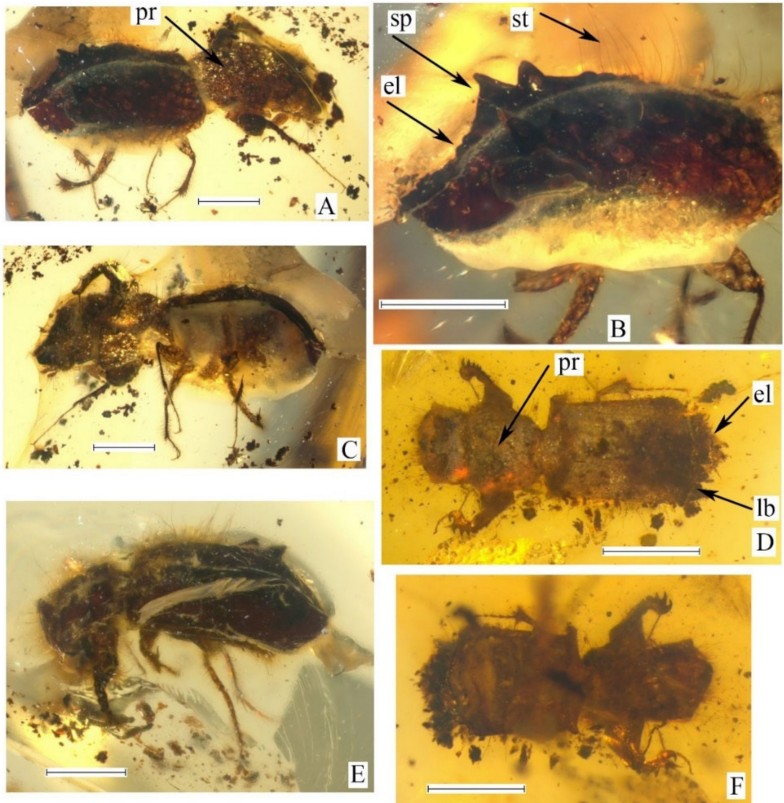

**Figure 3.** *Poinarinius* spp. in Myanmar amber: (**A**) *P. borowskii* sp. nov., holotype, female, ISEA MA2018/25, dorso-lateral view; (**B**) *P. borowskii* sp. nov., holotype, female, ISEA MA2018/25, elytra, dorsal view; (**C**) *P. borowskii* sp. nov., holotype, female, ISEA MA2018/25, ventral view; (**D**) *P. lesnei* sp. nov., holotype, female, ISEA MA2018/9, dorsal view; (**E**) *P. borowskii* sp. nov., paratype, female, ISEA MA2018/8, lateral view; (**F**) *P. lesnei* sp. nov., holotype, female, ISEA MA2018/9, ventral view. Scale bar 1.0 mm. Abbreviations: pr—pronotum, sp—spine, st—setae, lb—lobe, el—elytral declivity.

**Redescription**. Body red-brown, brown or black-brown, length 1.7–2.7mm. Integument covered with erect and semierect setae. Head slightly wider than the pronotum apical width. Frons convex, densely punctate, covered with rare erect setae. Vertex weakly impressed in the middle. First antennomere suboval, about 1.7 times as long as wide at the apex. Second antennomere conical, about 1.3 times as long as wide at the apex, about 0.6 times as long as and about 0.8 times as narrow as the first antennomere. Third and fourth antennomeres suboval. Third antennomere about 1.8 times as long as wide at the apex, about 0.6 times as long as and about 0.4 times as narrow as the second antennomere. Fourth antennomere equal to the third antennomere. Fifth antennomere wide-conical about 0.8 times as long as wide at the apex, about 0.6 times as long as and about 1.3 times as wide as the fourth antennomere. Sixth antennomere about 0.6 times as long as wide at the apex, about 3.0 times as long as and about 4.0 times as wide as the fifth antennomere. Seventh antennomere subequal to about 0.5 times as long as wide at the apex, slightly shorter and subequal in width to the sixth antennomere. Eighth antennomere about 0.7 times as long as wide at the apex, subequal in length, and 0.7 times as narrow as the seventh antennomere.

Pronotum 0.9–1.0 times as long as wide at the apex and in the middle, 1.0–1.1 times as long as wide at the base. Disk weakly flattened, densely punctate and longitudinally rugose. Elytra subcylindrical, 2.3–2.5 times as long as the pronotum, 1.8–2.3 times as long as wide at the base, 1.8–2.2 times as long as wide in the middle, 1.8–2.1 times as long as the apical fourth. Elytral intervals narrow, convex, equal in width or narrower than the striae. Striae deep, with large rounded punctation. Elytra with distinct elytral declivity, without lobes. Edges of elytral declivity serrated. Metaventrite about 2.5 times as long as the metacoxal cavity length. First ventrite slightly shorter than the metacoxal cavity length. Second ventrite slightly longer than the first ventrite. Third ventrite slightly shorter than the second ventrite. Fourth ventrite subequal to the third ventrite. Fifth ventrite slightly shorter than the fourth ventrite. Protarsi subequal in length to the protibiae. Mesotarsi about 1.5 times as long as the mesotibiae. Metatarsi about 1.3 times as long as the metatibiae. Protarsi: first tarsomere about 1.3 times as long as the second tarsomere; second tarsomere about 1.1 times as long as the third tarsomere; fourth tarsomere about 0.8 times as long as the third tarsomere; fifth tarsomere about 2.3 times as long as the fourth tarsomere, subequal to the third and fourth tarsomeres combined. Mesotarsi: first tarsomere about 1.8 times as long as the second tarsomere; second tarsomere subequal to the third tarsomere; fourth tarsomere about 0.8 times as long as the third tarsomere; fifth tarsomere about 2.3 times as long as the fourth tarsomere, subequal to the third and fourth tarsomeres combined. Metatarsi: first tarsomere about 1.6 times as long as second tarsomere; second tarsomere about 1.1 times as long as third tarsomere; fourth tarsomere about 0.8 times as long as third tarsomere; fifth tarsomere about 3.0 times as long as the fourth tarsomere, about 1.3 times as long as the third and fourth tarsomeres combined.

**Material examined**. Male (ISEA), no. MA2018/30; male (ISEA), no. MA2018/20; male (ISEA), no. MA2019/26; male (ISEA), no. MA2020/1; male (ISEA), no. MA2020/2; male (ISEA), no. MA2020/3; male (ISEA), no. MA2020/11; male (ISEA), no. MA2020/18; male (ISEA), no. MA2020/19; ex. (ISEA), no. MA2020/16; ex. (ISEA), no. MA2020/17; ex. (ISEA), no. MA2020/30; ex. (ISEA), no. MA2020/31; male (ISEA), no. MA 2018/24; (JHAC), no. JH2022/1; ex (JHAC), no. JH2022/2; ex. (JHAC), no. JH2022/3.

**Remarks.** This species is one of the most common Bostrichidae in Burmese amber. Peng et al. (2022) indicated 10-segmented antennae in the original description. We studied the antennae of our specimens, which were 8-segmented. It is likely that some antennomeres were torn during fossilization and Peng et al. (2022) mistakenly took them for different segments.

*Poinarinius zahradniki* Legalov & Háva, **sp. n.** (Figure 2G,H).

LSIDurn:lsid:zoobank.org:act:76AEDE38-90AC-4ED5-8F25-37279B695F09.

**Description**. Female. Body brown, length 3.3 mm. Head about 1.1 times as wide as the pronotum apical width. Frons convex, densely punctate, with tubercle in the middle, covered with sparse erect setae. Vertex with two tubercles. Pronotum about 1.3 times as long as wide at the apex, equal in wide in the middle, about 1.1 times as long as wide at the base. Disk weakly convex, densely and finely punctate. Elytra cylindrical, about 2.1 times as long as the pronotum, about 1.9 times as long as wide at the base, in the middle, and at the apical fourth. Sides of elytra with sparse, erect setae. Elytral intervals narrow, weakly convex, subequal in width to the striae. Striae deep. Elytra with distinct oblique elytral declivity, without lobes. Edges of elytral declivity even, quite high. Metaventrite about 3.2 times as long as the metacoxal cavity length. Protarsi subequal in length to the protibiae. Protarsi: first tarsomere about 1.6 times as long as the second tarsomere; second tarsomere subequal to the third tarsomere; fourth tarsomere about 0.8 times as long as the third tarsomere; fifth tarsomere about 2.0 times as long as the fourth tarsomere, slightly shorter than the third and fourth tarsomeres combined.

**Material examined**. Holotype—female (ISEA), no. MA2018/27.

**Etymology.** In honor of Petr Zahradník (Prague), specialist in Bostrichidae and Ptinidae.

**Comparison.** The new is similar to *P. aladelicatus* but differs in the larger body (3.3 mm) and even higher edges of the elytral declivity.

*Poinarinius burmaensis* Legalov, 2018.

*Poinarinius burmaensis* Legalov, 2018: 211–212, Figures 1–3.

**Material examined**. Holotype—female, (ISEA), no. MA2018/3; female (JHAC), JH2021/1; female (JHAC), no. JH2021; ex. (JHAC), no. JH2022.

**Remarks.** Description see in Legalov (2018).

*Poinarinius borowskii* Legalov & Háva, **sp. n.** (Figure 3A–C,E).

LSIDurn:lsid:zoobank.org:act:31607A3B-BEA8-4B43-9DF3-EC637F4BDDDF.

**Description**. Body brown, length 3.8–4.5 mm, covered with long quite dense erect setae. Frons flattened, densely punctate, covered with sparse erect setae. Pronotum about 1.1 times as long as wide at the apex, subequal in wide in the middle, about 1.3 times as long as wide at the base. Disk weakly convex, sparsely, and finely punctate. Elytra subcylindrical, about 2.6 times as long as the pronotum, about 2.4 times as long as wide at the base, 2.0 times as long as wide in the middle, and about 2.2 times as long as wide at the apical fourth. Disk and sides of elytra with erect setae. Elytral intervals wide, flat, about 5.0 times as wide as the striae. Striae weak. Elytra with weak elytral declivity and with double spines on the sides of elytral declivity. Protarsi about 1.1 times as long as the protibiae. Mesotarsi about 1.4 times as long as the mesotibiae. Protarsi: first tarsomere about 2.3 times as long as the second tarsomere; second tarsomere subequal to the third tarsomere; fourth tarsomere about 0.5 times as long as the third tarsomere; fifth tarsomere about 3.0 times as long as the fourth tarsomere, equal to the third and fourth tarsomeres combined. Mesotarsi: first tarsomere about 2.0 times as long as the second tarsomere; second tarsomere subequal to the third tarsomere; fourth tarsomere about 0.7 times as long as the third tarsomere; fifth tarsomere about 2.0 times as long as the fourth tarsomere, about 0.8 times as long as the third and fourth tarsomeres combined. Metatarsi: first tarsomere about 2.3 times as long as the second tarsomere; second tarsomere subequal to the third tarsomere; fourth tarsomere about 0.5 times as long as the third tarsomere; fifth tarsomere about 3.3 times as long as the fourth tarsomere, about 1.1 times as long as the third and fourth tarsomeres combined.

**Material examined**. Holotype—female (ISEA), no. MA2018/8. Paratypes: male (JHAC), no. JH2022/9; female (ISEA), no. MA2018/25.

**Etymology.** In honor of Jerzy Borowski (Warszawa), specialist in Bostrichidae and Ptinidae.

**Comparison.** The new is distinguished from *P. burmaensis* in the punctate pronotum. It differs from *P. cretaceus* **sp. nov.** in the elytra with double spines on the sides of the elytral declivity.

*Poinarinius lesnei* Legalov & Háva, **sp. n.** (Figure 3D,F and Figure 4F).

LSIDurn:lsid:zoobank.org:act:D15AFE9F-0574-44D4-BFF4-CD70EE9D3C99.

**Description**. Female. Body dark brown, length 3.1 mm, covered with long, erect setae. Head about 1.1 times as wide as the pronotum apical width. Frons weakly convex, densely punctate, with tubercle in middle, covered with quite dense erect setae. Vertex with two tubercles. Pronotum about 0.8 times as long as wide at the apex and in the middle, 0.9 times as long as wide at the base. Disk weakly flattened, finely punctate. Sides weakly rounded, with long erect setae. Elytra subcylindrical, about 2.5 times as long as the pronotum, about 1.7 times as long as wide at the base, about 1.5 times as long as wide in the middle, about 1.6 times as long as the apical fourth. Sides of elytra with long, erect setae. Elytral intervals flattened, punctate, 1.4–1.6 times as the striae. Striae deep. Elytra with double spines on the sides of the weak elytral declivity. Metaventrite 2.5 times as long as the metacoxal cavity length. First ventrite slightly longer than the metacoxal cavity length. Second ventrite equal to the first ventrite. Third ventrite about 1.2 times as long as the second ventrite. Protarsi about 1.3 times as long as the protibiae. Mesotarsi about 2.3 times as long as the mesotibiae. Metatarsi about 1.6 times as long as the metatibiae. Protarsi: first tarsomere 2.0 times as long as the second tarsomere; second tarsomere subequal to the third tarsomere; fourth tarsomere 0.7 times as long as the third tarsomere; fifth tarsomere about 2.3 times as long as the fourth tarsomere, slightly shorter than the third and fourth tarsomeres combined. Mesotarsi: first tarsomere 1.8 times as long as second tarsomere; second tarsomere about 1.1

times as long as third tarsomere; fourth tarsomere about 0.8 times as long as third tarsomere; fifth tarsomere about 2.7 times as long as fourth tarsomere, about 1.2 times as long as than third and fourth tarsomeres combined. Metatarsi: first tarsomere about 1.7 times as long as the second tarsomere; second tarsomere subequal to the third tarsomere; fourth tarsomere about 0.6 times as long as the third tarsomere; fifth tarsomere about 2.6 times as long as the fourth tarsomere, slightly shorter than the third and fourth tarsomeres combined.

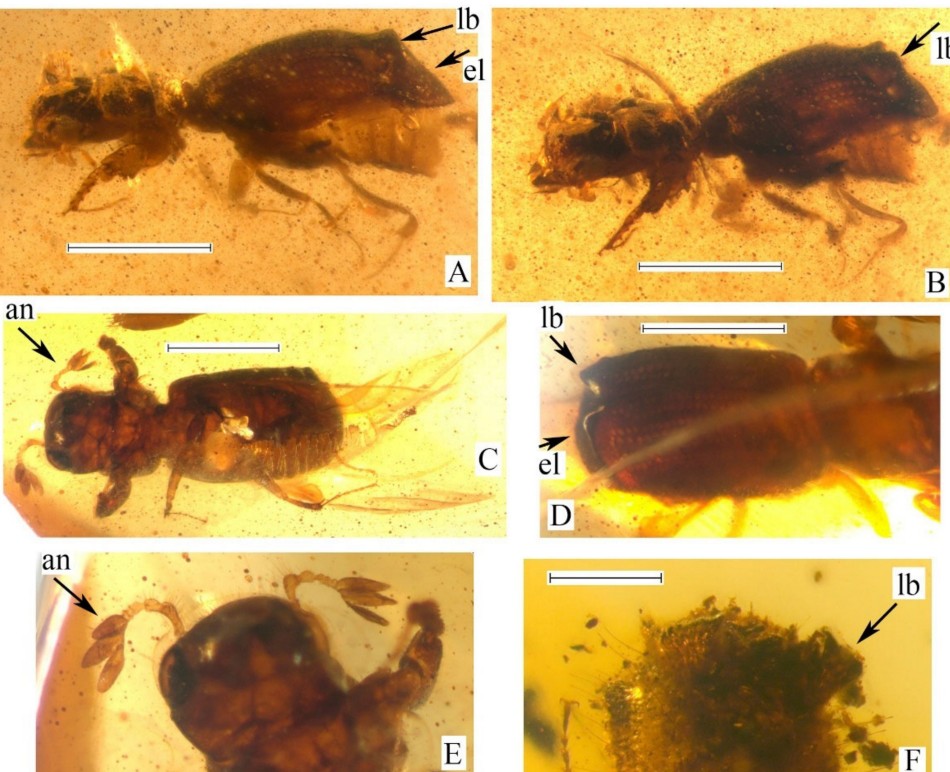

**Figure 4.** *Poinarinius* spp. in Myanmar amber: (**A**) *P. cretaceus* sp. nov., paratype, male, ISEA MA2020/5, lateral view; (**B**) *P. cretaceus* sp. nov., paratype, male, ISEA MA2020/5, dorso-lateral view; (**C**) *P. cretaceus* sp. nov., holotype, male, ISEA MA2018/4, ventral view; (**D**) *P. cretaceus* sp. nov., holotype, male, ISEA MA2018/4, elytra, dorso-lateral view; (**E**) P. cretaceus sp. nov., holotype, male, ISEA MA2018/4, antennae and head, ventral view; (**F**) *P. lesnei* sp. nov., holotype, female, ISEA MA2018/9, elytral cavity, dorsal view. Scale bar 1.0 mm. Abbreviations: lb—lobe, el—elytral declivity, an—antenna.

**Material examined**. Holotype—female (ISEA), no. MA2018/9; Paratypes: female (JHAC, no. JH2022/6; female (JHAC, no. JH2022/8.

**Etymology.** In memory dedicated to Pierre Lesne (*9.4.1871-†10.11.1949)—the famous expert of the family Bostrichidae.

**Comparison.** The new differs from *P. perkovskyi* **sp. nov.** in the elytra with two transverse lobes before the elytral declivity.

*Poinarinius cretaceus* Legalov & Háva, **sp. n.** (Figure 4A–E).

LSIDurn:lsid:zoobank.org:act:3479B170-66A1-4163-95AE-22893E9E1DCE.

**Description**. Male. Body red-brown, length 2.8–3.2 mm. Integument covered with erect and semierect long setae. Head about 1.2 times as wide as the pronotum apical width. Frons convex, sparsely punctate, covered with rare erect setae. Vertex weakly convex. First antennomere long conical, weakly curved, about 2.3 times as long as wide at the apex. Second and third antennomeres conical. Second antennomere about 1.4 times as long as wide at the apex, about 0.4 times as long as and about 0.7 times as narrow as the first antennomere. Third antennomere subequal in the length and width, about 0.6 times as long as and about 0.8 times as narrow as the second antennomere. Fourth and fifth

antennomeres are wide-conical. Fourth antennomere about 0.6 times as long as wide at the apex, about 0.6 times as long as and about 1.1 times as wide as the third antennomere. Fifth antennomere about 0.7 times as long as wide at the apex, about 1.2 times as long as and equal in width to the fourth antennomere. Sixth antennomere about 0.7 times as long as wide at the apex, about 2.7 times as long as and about 2.7 times as wide as the fifth antennomere. Seventh antennomere about 0.4 times as long as wide at the apex, equal in length and about 2.9 times as wide as the sixth antennomere. Eighth antennomere about 0.3 times as long as wide at the apex, slightly shorter, and slightly narrower than the seventh antennomere. Pronotum about 0.9 times as long as wide at the apex, in the middle, and at the base. Disk weakly convex, densely and finely punctate, with quite long setae. Elytra subcylindrical, 2.2–2.6 times as long as the pronotum, 1.6–2.0 times as long as wide at the base, about 1.6 times as long as wide in the middle, 1.6–1.7 times as long as the apical fourth. Elytral intervals narrow, convex, about 0.7 times as narrow as the striae. Striae deep, with large rounded punctation. Elytra with two longitudinal lobes before weak elytral declivity. Metaventrite about 1.8 times as long as the metacoxal cavity length. Metanepisterna about 3.8 times as long as wide in the middle. First ventrite about 0.6 times as long as the metacoxal cavity length. Second ventrite slightly longer than the first ventrite. Third ventrite slightly shorter than the second ventrite. Fourth ventrite about 1.6 times as long as the third ventrite. Fifth ventrite about 0.9 times as long as the fourth ventrite. Metabibial long spur about 2.5 times as long as the short one. Protarsi about subequal in length to the protibiae. Mesotarsi about 1.6 times as long as the mesotibiae. Metatarsi about 1.5 times as long as the metatibiae. Protarsi: first tarsomere about 1.6 times as long as the second tarsomere; second tarsomere about 1.3 times as long as the third tarsomere; fourth tarsomere about 0.5 times as long as the third tarsomere; fifth tarsomere about 3.0 times as long as the fourth tarsomere, subequal to the third and fourth tarsomeres combined. Mesotarsi: first tarsomere about 2.0 times as long as the second tarsomere; second tarsomere about 1.2 times as long as the third tarsomere; fourth tarsomere about 0.8 times as long as the third tarsomere; fifth tarsomere about 2.3 times as long as the fourth tarsomere, subequal to the third and fourth tarsomeres combined. Metatarsi: first tarsomere about 2.1 times as long as the second tarsomere; second tarsomere about 1.2 times as long as the third tarsomere; fourth tarsomere about 0.7 times as long as the third tarsomere; fifth tarsomere about 2.8 times as long as the fourth tarsomere, slightly longer than the third and fourth tarsomeres combined.

**Material examined**. Holotype—male (ISEA), no. MA2018/4. Paratypes: male (ISEA), no. MA2020/5; male (ISEA), no. MA2020/33.

**Etymology.** From the Cretaceous period.

**Comparison.** The new is similar to *P. borowskii* **sp. nov.**, but differs in the elytra with two longitudinal lobes before the elytral declivity.

**Key to species of the genus** *Poinarinius.*
1. Elytra lacking distinct elytral declivity and lobes (Figure 1A,B,E–G) . . . . . . . . . . . . . . . . . . . 2
— Elytra with more or less distinct elytral declivity (Figure 2A–D,F–H, Figure 3A,B,D,E, and Figure 4A,B,D,F) . . . . . . . . . . . . . . . . . . . . . . . . . . . . . . . . . . . . . . . . . . . . . . . . . . . . . . . . . . . . . . . . . . . . . . . . . . . . . . . . . ..3
2. Pronotum punctate (Figure 1A,B,E,F) . . . . . . . . . . . . . . . . . . . . . . . . . . ..*P. aristovi* **sp. nov.**
— Pronotum transverse rugose (Figure 1G) . . . . . . . . . . . . . . . . . . . . . *P. antonkozlovi* **sp. nov.**
3. Elytral declivity almost vertical (Figure 2A, Figure 3D, and Figure 4F) . . . . . . . . . . . . . . . . . . . . . . . . . . . . . ..4
— Elytral declivity oblique (Figure 2B–D,F–H, Figure 3A,B,E, and Figure 4A,B,D) . . . . . . . . . . . . . . . . . 5

4. Elytra with two transverse lobes before elytral declivity (Figures 3D and 4F) . . . .. . . . .
. . . . . . . . . . . . . . . . . . . . . . . . . . . . . . . . . . . . . . . . . . . . . . . . . . . . . . . . . . . . . . . . . . . .
*P. lesnei* **sp. nov.**
— Elytra with two spines before elytral declivity (Figure 2A) . . . ..*P. perkovskyi* **sp. nov.**
5. Elytra with distinct elytral declivity, without lobes (Figure 2B–D,F–H) . . . . . . . . . . . . .6
— Elytra with weak elytral declivity, but with lobes or spines (Figure 3A,B,E and Figure 4A,B,D)
. . . . . . . . . . . . . . . . . . . . . . . . . . . . . . . . . . . . . . . . . . . . . . . . . . . . . . . . . . . . . . . . . .
. . . . . . . . . . . . . . . . . . 7
6. Body smaller (1.7–2.7 mm). Edges of elytral declivity serrated, lower (Figure 2B–D,F) . . .
. . . . . . . . . . . . . . . . . . . . . . . . . . . . . . . . . . . . . . . . . . . . . . . . . . . . . . . . . . . . . . . . . . .
. . . ..*P. aladelicatus*
— Body larger (3.3 mm). Edges of elytral declivity even, higher (Figure 2G,H) . . . . . . . . .
. . . . . . . . . . . . . . . . . . . . . . . . . . . . . . . . . . . . . . . . . . . . . . . . . . . . . . . .*P. zahradniki*
**sp. nov.**
7. Pronotum transverse rugose. Elytra with longitudinal lobes on the sides of very weak
elytral declivity . . . . . . . . . . . . . . . . . . . . . . . . . . . . . . . . . . . . . . . . . . . . . . . . . . . . . . .
. . . . . . *P. burmaensis*
— Pronotum punctate . . . . . . . . . . . . . . . . . . . . . . . . . . . . . . . . . . . . . . . . . . . . . . . . . . . .
. . . . . . . . . . . . . . . ..8
8. Elytra with double spines on sides of elytral declivity (Figure 3A,B,E) . . . . . . . . . . . .
. . . . . . . . . . . . . . . . . . . . . . . . . . . . . . . . . . . . . . . . . . . . . . . . . . . . . . . . . *P. borowskii* **sp.**
**nov.**
—Elytra with two longitudinal lobes before elytral declivity (Figure 4A,B,D) . . . . . . . . . . .
. . . . . . . . . . . . . . . . . . . . . . . . . . . . . . . . . . . . . . . . . . . . . . . . . . . . . . . . . . *P. cretaceus* **sp.**
**nov.**

**List of the fossil Bostrichidae**
Polycaoninae.
Genus *Cretolgus* Legalov & Háva, 2020.
*C. minimus* Legalov & Háva, 2020—Burmese amber, 1 ex. [9].
Dinoderinae.
Genus *Stephanopachys* Waterhouse, 1888.
*S. vetus* Peris, Delclòs et Perrichot, 2014—French amber, 1 ex. [7].
*S. electron* Zahradník & Háva, 2015—Baltic amber, 1 ex. [11].
*S. ambericus* Zahradník & Háva, 2015—Baltic amber, 1 ex. [11].
Genus *Dinoderus* Stephens, 1830.
*D. cuneicollis* Wickham, 1913—Florissant, 1 ex. [14].
Genus? *Rhizopertha* Stephens, 1830.
*Rh.* sp.—Baltic amber, 3 ex. [40].
Genus ? *Prostephanus* Lesne, 1898.
*P.* sp.—Mexican amber, 1 ex. [16].
Alitrepaninae Peng, Jiang, Engel & Wang, 2022.
Genus *Poinarinius* Legalov, 2018.
*P. aladelicatus* (Peng, Jiang, Engel & Wang, 2022)—Burmese amber, 19 ex. [10] and
presented data.
*P. antonkozlovi* Legalov & Háva, **sp. n.**—Burmese amber, 3 ex.
*P. aristovi* Legalov & Háva, **sp. n.**—Burmese amber, 19 ex.
*P. borowskii* Legalov & Háva, **sp. n.**—Burmese amber, 3 ex.
*P. burmaensis* Legalov, 2018—Burmese amber, 4 ex.
*P. cretaceus* Legalov & Háva, **sp. n.**—Burmese amber, 3 ex.
*P. lesnei* Legalov & Háva, **sp. n.**—Burmese amber, 3 ex.
*P. perkovskyi* Legalov & Háva, **sp. n.**—Burmese amber, 1 ex.
*P. zahradniki* Legalov & Háva, **sp. n.**—Burmese amber, 1 ex.
Lyctinae.
Genus?*Lyctus* Fabricius, 1792.

*L.* sp.—Baltic amber? ex. [41].
Bostrichinae.
Bostrichini.
Genus ? *Bostrichus* O.F. Mueller, 1764.
*B.* sp.—Baltic amber? ex. [41–43].
Genus *Amphicerus* LeConte, 1861.
*A. sublevis* Wickham, 1914—Florissant, 1 ex. [15].
Genus *Discoclavata* Poinar, 2013.
*D. dominicana* Poinar, 2013—Dominican amber, 1 ex. [17].
Apatini.
Genus?*Apate* Fabricius 1775.
*A.* sp.—Baltic amber, ? ex. [42,43].
Genus *Protapate* Wickham, 1912.
*P. contorta* Wickham, 1912—Florissant, 1 ex. [13].
Xyloperthini.
Genus *Xylobiops* Casey, 1898.
*X. lacustre* Wickham, 1912—Florissant, 2 ex. [13,14].

## 4. Discussion

The family Bostirichidae is represented by nine subfamilies, eight of which are present in the recent fauna [6,44], and one is extinct [10]. Molecular evidence suggests that the family appeared at the beginning of the late Jurassic [45], but Bostrichidae are recorded much later in the fossil history. The first subfamily Dinoderinae to appear in the fossil record is known from the Aptian of France [7] and found in the late Eocene of Europe [11,40] and North America [14], and has also been recorded from Miocene Mexican amber [16]. The subfamily Alitrepaninae is described only from Burmese amber [10]. The first record of the subfamily Polycaoninae is also from the Cenomanian of Myanmar [9]. Undescribed Lyctinae are reported from Baltic amber [41,46]. The earliest Bostrichinae are described from the terminal Eocene of the USA (tribes Apatini, Bostrichini, and Xyloperthini [13,15], but also recorded for the late Eocene of Europe [41–43]. The tribe Bostrichini was also found in Dominican amber [17]. Undescribed species of the tribe Apatini are known from Baltic amber [42,43]. The image of a beetle identified as Psoinae [46] refers to Eucnemidae. Representatives of the subfamilies Dysidinae, Euderinae, Endecatominae, and Psoinae of Bostrichidae have no fossil record [8,9,12]. Thus, one subfamily is known from the Early Cretaceous, and two subfamilies from the late Cretaceous. Two subfamilies appear in the fossil record in the late Eocene. It should be noted that the Bostrichid genera are given from Baltic amber according to the determinations of the early 19th century [41–43] and require confirmation.

Cretaceous and Eocene ambers with Bostrichidae were formed from resins of gymnosperms (probably Araucariaceae first, and Sciadopitys second), and Miocene from angiosperms (*Hymenaea*). In total, about 80 specimens of fossil bostirichids are known, and the vast majority of them are from amber. Five specimens of four species are represented by impressions. Almost 70% of Bostrichidae are found in Burmese amber (Figure 5). Despite the vast amount of material on beetles from Baltic amber, descriptions on auger beetles are rare (Figure 5). Hieke and Pietrzeniuk [40] wrote that there are only nine specimens available in the collections of the Museum of Natural History in Berlin. The exact number of Bostrichid remains in Baltic amber is not known, but it can be estimated as more than ten specimens. Five specimens of Bostirichidae are found in the Florissant [13–15]. Auger beetles are rare in Miocene amber.

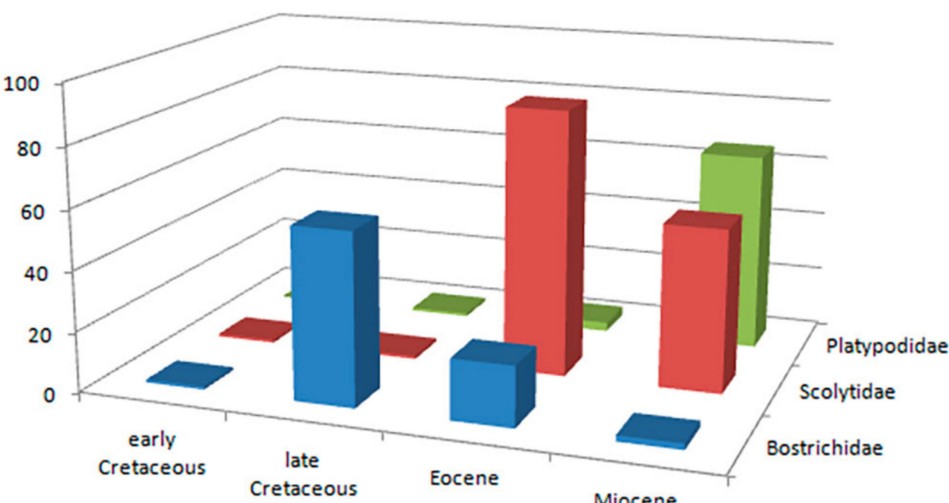

**Figure 5.** Number of auger, bark, and ambrosia beetles in the fossil record.

It is possible to make a correlation with other ecologically similar groups—bark (Scolytidae) and ambrosia beetles (Platypodidae). These groups are considered as independent families [23,27–29,38,47–52] or as subfamilies of the family Curculionidae [53–58]. The oldest bark beetles [21] and Bostrichidae are represented by single specimens from the Early Cretaceous (Figures 5 and 6). At the beginning of the late Cretaceous, bostrichids had greater diversity and maximum abundance, while bark and ambrosia beetles occurred singly (Figure 5) [22,23]. The situation changed in the Paleogene. The auger beetles probably increased their species diversity, but their numbers fell, while there was a significant increase in the diversity and abundance of bark beetles (Figures 5 and 6) [24–30,59], which were mainly associated with gymnosperms. Platypodids were also single [28,30]. The situation again changed in the Neogene. The ambrosia beetles increased the number of species and dominated in numbers [31–36], while bark beetles, with a rather high diversity (but lower than in the Eocene) (Figures 5 and 6) [33,37,38], faded into the background in tropic forest ecosystems. This change in diversity and abundance can be explained by the replacement of amber-producing trees from angiosperms to gymnosperms, which are well-colonized by ambrosia beetles. Finds of auger beetles became rare, and a similar situation persists today. The bark ambrosia beetles dominated the tropics, and the auger beetles had a much lower species richness and abundance than these groups. The auger and ambrosia beetles were quite rare in temperate latitudes, when bark beetles reached a huge number and a fairly large number of species.

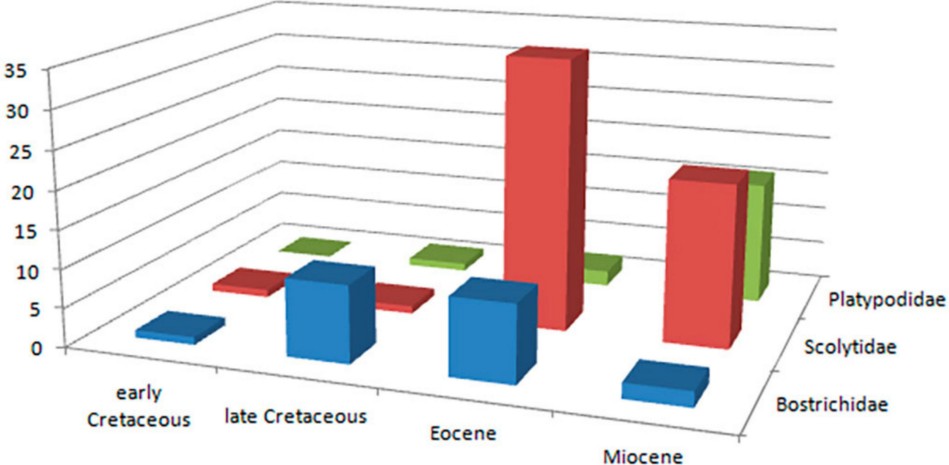

**Figure 6.** Species diversity of the auger, bark, and ambrosia beetles in the fossil record.

## 5. Conclusions

The auger beetles probably formed in the first half of the Early Cretaceous since a representative of the modern genus was found in the Aptian. They reached a high species diversity by the beginning of the late Cretaceous. Diversification at the level of tribes and genera likely occurred in the Paleogene. The subfamily Alitrepaninae that dominated the mid-Cretaceous of Asia was not found in other Cenomanian localities. Bostrichidae occupied the ecological niche of the bark and ambrosia beetles in the mid-Cretaceous forests. As the abundance and diversity of these groups increased, the role of the auger beetles in forest ecosystems decreased.

**Author Contributions:** A.A.L. and J.H. designed the study, prepared new species descriptions and their systematic placement, drafted the manuscript and contributed to the writing and discussion, read and agreed to the published version of the manuscript. A.A.L. prepared new species plates. All authors have read and agreed to the published version of the manuscript.

**Funding:** This research received no external funding.

**Institutional Review Board Statement:** Not applicable.

**Informed Consent Statement:** Not applicable.

**Data Availability Statement:** The specimens were deposited at the Institute of Systematics and Ecology of Animals, Siberian Branch, Russian Academy of Sciences (Novosibirsk) and Jiří Háva Private Entomological Laboratory & Collection, Únětice u Prahy, Prague-West, Czech Republic.

**Conflicts of Interest:** The authors declare no conflict of interest.

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
