# Peer review of "Diversity of Auger Beetles (Coleoptera: Bostrichidae) in the Mid-Cretaceous Forests with Description of Seven New Species"

_diversity, doi:10.3390/d14121114_

Round 1

Reviewer 1 Report

The manuscript describes new species of Bostrichidae. The topic is relevant. However, it is not well worked. The introduction is highly-saving. For example, it needs to be improved by adding an introduction to the subfamily and genera treated in the descriptions.

In the introduction, systematics notes and discussion, assumptions like “Bostrichidae is the most common beetle in Burmese amber” or “the structure of the apex of the elytra can be different” are presented without references. The ecological discussion is based on the first assumption, so a reference for such affirmations is essential. The variability of morphological characters needs to be properly discussed to synonymize a genus.

The authors claim that Bostrichidae makes 70% of the Burmese amber fauna with described species and their material. However, this information is hugely biased since the authors focus their work on this family; they put together an extensive collection.

The conclusions mentioned in the manuscript were already discussed in Peri et al. (2021) (https://doi.org/10.1111/brv.12763). However, this citation, e.g., is not mentioned.

The sources for Figures 5 and 6 are not mentioned.

A comparison between amber from the Cretaceous, Eocene, and Miocene without a differentiation of the origin makes little sense. The Angiosperms trees produced only a few amber deposits during the Eocene and not one during the Cretaceous. The same is also true for the Miocene and the gymnosperms.

Platypodinae and Scolytinae belong to the family Curculionidae and are subfamilies, not families.

Author Response

We thank the reviewer for the comments and corrected the manuscript according to this comments. With best wishes, Andrei and Jiri

Answers: 

The manuscript describes new species of Bostrichidae. The topic is relevant. However, it is not well worked. The introduction is highly-saving. For example, it needs to be improved by adding an introduction to the subfamily and genera treated in the descriptions.

- We have expanded the introduction.

In the introduction, systematics notes and discussion, assumptions like “Bostrichidae is the most common beetle in Burmese amber” or “the structure of the apex of the elytra can be different” are presented without references. The ecological discussion is based on the first assumption, so a reference for such affirmations is essential. The variability of morphological characters needs to be properly discussed to synonymize a genus.

- We have expanded the rationale for synonymy. Variation is important across species. Here, the volume of characters sufficient for recognition of a single genus is more correct.

In the introduction, systematics notes and discussion, assumptions like “Bostrichidae is the most common beetle in Burmese amber” or “the structure of the apex of the elytra can be different” are presented without references. The ecological discussion is based on the first assumption, so a reference for such affirmations is essential. The variability of morphological characters needs to be properly discussed to synonymize a genus.

- We have shown the most important features that determine the scope of the genus.

The authors claim that Bostrichidae makes 70% of the Burmese amber fauna with described species and their material. However, this information is hugely biased since the authors focus their work on this family; they put together an extensive collection.

- No, we are stating that Burmese Bostrichidae make up 70% of all known fossil bostrichids, not all Burmese amber Coleoptera.

The conclusions mentioned in the manuscript were already discussed in Peri et al. (2021) (https://doi.org/10.1111/brv.12763). However, this citation, e.g., is not mentioned.

- The conclusions of this work are not identical to those of Peris et al. 2021, but I have added a link to this work.

The sources for Figures 5 and 6 are not mentioned. - it was done

A comparison between amber from the Cretaceous, Eocene, and Miocene without a differentiation of the origin makes little sense. The Angiosperms trees produced only a few amber deposits during the Eocene and not one during the Cretaceous. The same is also true for the Miocene and the gymnosperms.

– It is not ambers that are compared, but faunas in inclusions, and amber-bearing trees are not important in this comparison, but the change in the diversity of beetles is important.

Platypodinae and Scolytinae belong to the family Curculionidae and are subfamilies, not families. - No, these are separate families, see Morimoto and Kojima (2004) or Legalov 2015

Reviewer 2 Report

This is an interesting manuscript dealing with a diversity of auger beetles (Coleoptera; Bostrichidae from the Cretaceous Burmese amber. This study enriches our knowledge of the family Bostrichidae during the mid Cretaceous in east Asia. The ms in the current form needs some significant improvement. Most new taxa (new species) described in this manuscript lack detailed images that show the diagnostic characters for each new taxon. Also, some given images are too blurry (e.g., Figs 1B,H, 2E, 3F, 4D) and cannot provide convincing evidence. More importantly, considering taphonomic deformation is very common in the bioinclusions of Burmese amber, the description for each species needs more serious justification and take into full consideration of the taphonomic artefacts. Some characters, such as the body length, are not very suitable for the key. In other words, a short remark/comparative note is required for new taxa.

Small points:

1. lower/early Cretaceous should be Lower/Early Cretaceous;

2. In many places, hyphen should be replaced by en dash.

Author Response

We thank the reviewer for the comments and corrected the manuscript according to this comments. With best wishes, Andrei and Jiri

Answers:

This is an interesting manuscript dealing with a diversity of auger beetles (Coleoptera; Bostrichidae from the Cretaceous Burmese amber. This study enriches our knowledge of the family Bostrichidae during the mid Cretaceous in east Asia. The ms in the current form needs some significant improvement. Most new taxa (new species) described in this manuscript lack detailed images that show the diagnostic characters for each new taxon. Also, some given images are too blurry (e.g., Figs 1B,H, 2E, 3F, 4D) and cannot provide convincing evidence. More importantly, considering taphonomic deformation is very common in the bioinclusions of Burmese amber, the description for each species needs more serious justification and take into full consideration of the taphonomic artefacts. Some characters, such as the body length, are not very suitable for the key. In other words, a short remark/comparative note is required for new taxa.

– We added comparisons and depicted important signs for the arrows in the illustrations

Small points:

  1. lower/early Cretaceous should be Lower/Early Cretaceous; - it was done

Reviewer 3 Report

1.Abstract: 

........ that Bostrichidae played the ecological role of the bark and 14 ambrosia beetles in the Mesozoic. 

please make it clear for this sentence.

2. Abstract: Seven new species, Paristovi sp. nov.

Complete genus name when it appears for the first time in the MS. 

3. Introduction 

Line 23-24: The beetles are among the main inhabitants of various forests, where they are associated with both living trees and dead decaying wood. 

Which beetles? delimit them. Sure not all. 

4. Line 247 (page 7): Poinarinius aladelicatum (Peng, Jiang, Engel & Wang, 2022), comb. nov. 

Poinarinius aladelicatus? Check if the ending should be changed according the gender of the genus name newly combined.

5. Lines 293-297 (page 8): Remarks. This species is one of the most common Bostrichidae in Burmese amber. 293 Peng et al. (2022) indicated 10-segmented antennae in the original description. We stud-294 ied the antennae of our specimens. They are 8-segmented. Probably some antennomeres 295 were torn during fossilization and Peng et al. (2022) mistakenly took them for different 296 segments.

Please check the holotype or original photos in the paper that the name was originally published. Remarks did not give enough support for this synonymizing.

Author Response

We thank the reviewer for the comments and corrected the manuscript according to this comments. With best wishes, Andrei and Jiri

Answers:

1.Abstract:

........ that Bostrichidae played the ecological role of the bark and 14 ambrosia beetles in the Mesozoic. please make it clear for this sentence. - it was done

  1. Abstract: Seven new species, P. aristovi sp. nov.,

Complete genus name when it appears for the first time in the MS. - it was done

  1. Introduction

Line 23-24: The beetles are among the main inhabitants of various forests, where they are associated with both living trees and dead decaying wood.

Which beetles? delimit them. Sure not all. - it was done

  1. Line 247 (page 7): Poinarinius aladelicatum (Peng, Jiang, Engel & Wang, 2022), comb. nov.

Poinarinius aladelicatus? Check if the ending should be changed according the gender of the genus name newly combined. - it was done

  1. Lines 293-297 (page 8): Remarks. This species is one of the most common Bostrichidae in Burmese amber. 293 Peng et al. (2022) indicated 10-segmented antennae in the original description. We stud-294 ied the antennae of our specimens. They are 8-segmented. Probably some antennomeres 295 were torn during fossilization and Peng et al. (2022) mistakenly took them for different 296 segments.

Please check the holotype or original photos in the paper that the name was originally published. Remarks did not give enough support for this synonymizing.

- There is no need to study the holotype as this species (aladelicatus) is easily identified. We re-examined our materials under a high magnification microscope. The beetles are with 8-segmented antennae (scape+4-segmented funicle+3 segmented club), as in aladelicatus as in other species of this genus.

Reviewer 4 Report

An interesting work that gives new information about the origin and diversity of auger beetles. The work is based on nice material, and therefore its main conclusions are clear and substantiated. Descriptions and diagnoses of new species are well written. Nevertheless, in general English language and style of the MS requires serious improvement.    

Also, the part of the first paragraph of Discussion section (lines 510-523) provides valuable and detailed information about the taxa in question and objects of the study, i.e. "introduction to the topic", NOT the real discussion. Thus, it is better to move this part to Introduction section. 

To conclude, my opinion at this stage that the MS can be accepted for publication after minor revision and English editing and style correction. 

Author Response

We thank the reviewer for the comments. With best wishes, Andrei and Jiri

Answer:

We would like to leave this part for discussion as it leads logically to the analysis of bostrichid diversity in the fossil record.

Round 2

Reviewer 1 Report

The title says that Bostrichidae was a characteristic feature of the mid Cretaceous forest. The abstract and the conclusion claims that the role of bark and ambrosia beetles was played by the Bostrichidae. However, this is on the basis of selected amber pieces with this kind of inclusion (the sample is not unbiased). Thus, the conclusion is not well supported.

The answer: “It is not ambers that are compared, but faunas in inclusions, and amber-bearing trees are not important in this comparison, but the change in the diversity of beetles is important.” Is not understandable for me because the ecological role has much to do with amber-bearing trees.

My strong suggestion is to change the title and the discussion around the topic “ecological role” since it is much speculation, and let it be much more conservative.

The sentence “The specimens of Bostrichidae are among the common beetles in Burmese amber.” Is still without reference. Moreover, the publication mentioned after (Peng et al 2022) explains that this kind of beetle is uncommon.

The introduction is still highly-saving. 

Platypodinae and Scolytinae belong to the family Curculionidae and are subfamilies, not families. This has been extensively discussed, for example here (10.3897/zookeys.439.8391).

Author Response

We thank the reviewer for his comments, although our opinions do not coincide, we tried to make any corrections that were possible.

With best wishes, Andrei Legalov and Jiri Hava

The title says that Bostrichidae was a characteristic feature of the mid Cretaceous forest. The abstract and the conclusion claims that the role of bark and ambrosia beetles was played by the Bostrichidae. However, this is on the basis of selected amber pieces with this kind of inclusion (the sample is not unbiased). Thus, the conclusion is not well supported.

- W do not agree, since the selection is very objective, we represent the number of ambers with Bostrichidae and other Coleoptera offered for sale, but we have changed the name according to the remark.

The answer: “It is not ambers that are compared, but faunas in inclusions, and amber-bearing trees are not important in this comparison, but the change in the diversity of beetles is important.” Is not understandable for me because the ecological role has much to do with amber-bearing trees.

- Naturally, the ecological role is associated with trees, but this paragraph is about changing the diversity of the group.

My strong suggestion is to change the title and the discussion around the topic “ecological role” since it is much speculation, and let it be much more conservative.

- W have made corrections to the title and phrase “ecological role”, but we will note that everything in paleontology is based on speculation since the objects have long died out.

The sentence “The specimens of Bostrichidae are among the common beetles in Burmese amber.” Is still without reference. Moreover, the publication mentioned after (Peng et al 2022) explains that this kind of beetle is uncommon.

- By the fact that these are our observations of the materials entering the stores and online auctions. Actually our article is a confirmation that these beetles are common in Burmese amber.

Platypodinae and Scolytinae belong to the family Curculionidae and are subfamilies, not families. This has been extensively discussed, for example here (10.3897/zookeys.439.8391).

- We think otherwise and consider these groups as separate families descended from the Mesozoic Ithyceridae.

Round 3

Reviewer 1 Report

- Naturally, the ecological role is associated with trees, but this paragraph is about changing the diversity of the group.

Today, we would not compare the diversity from two different regions if, in each region, different collection methods (different kinds of resins) were used, and insects were collected in different environments (Gymnosperm forest vs. Angiosperm forest). For this reason, a change in the diversity and the ecological role can’t be explained just by comparing the described diversity in amber from the Cretaceous, Eocene, and Miocene. I strongly recommend to consider this aspect in the discussion.

- W have made corrections to the title and phrase “ecological role”, but we will note that everything in paleontology is based on speculation since the objects have long died out.

Statistical methods and the ecology of living specimens provide strong evidence to support and understand environments in the past. Paleontology is not just based on speculation

- By the fact that these are our observations of the materials entering the stores and online auctions. Actually, our article is a confirmation that these beetles are common in Burmese amber.

Bostrichidae and other Coleoptera offered for sale can’t be the basis for a scientific dataset. If we are looking for Bostrichidae in the market, we will find them, and the dataset is then biased. In the introduction, the supposition that Bostrichidae is common in amber because the author finds it on the market is not a statement. That these beetles are common on the online markets confirms only that they can be found on the online market.

- We think otherwise and consider these groups as separate families descended from the Mesozoic Ithyceridae.

Please add the other point of view

Author Response

We thank the referee for his comments, which improved our work.

Andrei and Jiri

Reviewer's comments:

“Today, we would not compare the diversity from two different regions if, in each region, different collection methods (different kinds of resins) were used, and insects were collected in different environments (Gymnosperm forest vs. Angiosperm forest). For this reason, a change in the diversity and the ecological role can’t be explained just by comparing the described diversity in amber from the Cretaceous, Eocene, and Miocene. I strongly recommend to consider this aspect in the discussion.”

- We've done it.

“Statistical methods and the ecology of living specimens provide strong evidence to support and understand environments in the past. Paleontology is not just based on speculation”

- Naturally, this is so, We use it all the time in my work, but this is just an assumption, nothing more.

“Bostrichidae and other Coleoptera offered for sale can’t be the basis for a scientific dataset. If we are looking for Bostrichidae in the market, we will find them, and the dataset is then biased. In the introduction, the supposition that Bostrichidae is common in amber because the author finds it on the market is not a statement. That these beetles are common on the online markets confirms only that they can be found on the online market.”

- Absolutely disagree with this statement. Almost all of the mined material is exhibited on online markets, and what museums and private collectors acquire from it, i.e., gets into the collections. is the selected material. Therefore, online sales give a more objective picture than museum collections. This applies primarily to Burmese and Baltic amber.

“Please add the other point of view”

- We added that there is another point of view.
